# Maximally Expressive Graph Neural Networks for Outerplanar Graphs

**Franka Bause**[*1,2], **Fabian Jogl**[*3,4], **Patrick Indri**[3], **Tamara Drucks**[3], **David Penz**[3,5],
**Nils M. Kriege**[1,6], **Thomas Gärtner**[3], **Pascal Welke**[3], **Maximilian Thiessen**[3]

[1] *Faculty of Computer Science, University of Vienna, Vienna, Austria*

[2] *UniVie Doctoral School Computer Science, University of Vienna, Vienna, Austria*

[3] *Machine Learning Research Unit, TU Wien, Vienna, Austria*

[4] *Center for Artificial Intelligence and Machine Learning, Vienna, Austria*

[5] *Multimedia Mining and Search, Johannes Kepler University Linz, Linz, Austria*

[6] *Research Network Data Science, University of Vienna, Vienna, Austria*

{firstname.lastname}@{univie, tuwien}.ac.at

*Reviewed on OpenReview:* *https://openreview.net/forum?id=XxbQAsxrRC*

## Abstract

We propose a *linear time* graph transformation that enables the Weisfeiler-Leman (WL) algorithm and message passing graph neural networks (MPNNs) to be maximally expressive on *outerplanar* graphs. Our approach is motivated by the fact that most pharmaceutical molecules correspond to outerplanar graphs. Existing research predominantly enhances the expressivity of graph neural networks without specific graph families in mind. This often leads to methods that are impractical due to their computational complexity. In contrast, the restriction to outerplanar graphs enables us to encode the Hamiltonian cycle of each biconnected component in linear time. As the main contribution of the paper we prove that our method achieves maximum expressivity on outerplanar graphs. Experiments confirm that our graph transformation improves the predictive performance of MPNNs on molecular benchmark datasets at negligible computational overhead.

## 1 Introduction

We study graph neural networks (GNNs) for the class of outerplanar graphs and devise a linear time preprocessing step that message passing graph neural networks (MPNNs) to distinguish all non-isomorphic outerplanar graph. Morris et al. (2019) and Xu et al. (2019) showed that MPNNs have limited *expressivity*, i.e., there exist non-isomorphic graphs on which each MPNN produces the same embedding. Such graphs exist even within the restricted class of outerplanar graphs (see Figure 1). This led to the development of GNNs that are more expressive than MPNNs, often called *higher-order* GNNs. However, the increase in expressivity usually comes with a significant increase in computational complexity. For example, *k*-GNNs (Morris et al., 2019) have a time complexity of $\Omega(|V|^k)$. Other higher-order GNNs count pattern graphs such as cliques (Bodnar et al., 2021b), cycles (Bodnar et al., 2021a;b), and general subgraphs (Bouritsas et al., 2022), which can take time exponential in the pattern size. However, for certain domains of interest, the graph structure can be exploited to build efficient higher-order GNNs. In this work, we focus on the pharmaceutical domain and on graphs that represent molecules. Over 92% to 97% of the graphs in widely used benchmark datasets in this domain are *outerplanar* (see Table 1). The properties of outerplanar graphs have been exploited by algorithms for graph mining (Horváth et al., 2010) and molecular similarity computation (Schietgat et al., 2013; Droschinsky et al., 2017), but not yet for GNNs. We focus on this

---

*Equal contribution.

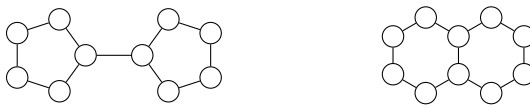

Figure 1: The molecules bicyclopentyl (left) and decalin (right) which correspond to outerplanar graphs that cannot be distinguished by MPNNs and 1-WL.

class of graphs and devise a linear time transformation that allows MPNNs to become maximally expressive on outerplanar graphs. This implies that, in principle, our architecture can solve any learning task on outerplanar graphs. While this does not mean that any given maximally expressive model will perform well in practice, our experiments show that our proposed transformation improves the predictive performance of several GNN architectures on multiple benchmark learning tasks.

We propose to decompose outerplanar graphs into biconnected outerplanar components and trees. Using the fact that each biconnected outerplanar component has a unique Hamiltonian cycle that can be computed in linear time, we split each component into two graphs corresponding to the directed Hamiltonian cycles, and prove that MPNNs are maximally expressive on biconnected outerplanar graphs transformed in this way. Taking advantage of the well-known fact that MPNNs are maximally expressive on labeled trees (Arvind et al., 2015; Kiefer, 2020), we extend our result to a linear time graph transformation called *Cyclic Adjacency Transform* (CAT) that works on all outerplanar graphs. We benchmark CAT with common MPNNs on a variety of molecular graph benchmarks and show that CAT consistently boosts the performance of MPNNs.

**Main contributions.** We propose CAT, a linear time graph transformation that renders MPNNs maximally expressive on outerplanar graphs. Experimentally, CAT consistently improves the performance of MPNNs with little increase in runtime.

## 2 Discussion and Related Work

Since the expressivity of MPNNs is bounded by the 1-WL algorithm (Morris et al., 2019; Xu et al., 2019), any pair of non-isomorphic graphs that cannot be distinguished by 1-WL will get mapped to the same embedding by any given MPNN. One such pair of graphs are the molecules decalin and bicyclopentyl (see Fig. 1). As these two graphs are outerplanar, it follows that MPNNs are not sufficiently expressive for outerplanar graphs. Furthermore, in the graph mining community it is well known that many pharmaceutical molecules are outerplanar (Horváth et al., 2006; Horváth & Ramon, 2010). Outerplanarity has also been discussed in the context of reconstruction with GNNs (Cotta et al., 2021). This motivates the need for GNNs that are highly expressive on outerplanar graphs. Outerplanar graphs have treewidth at most two (Bodlaender, 1998), and Kiefer (2020) showed that 3-WL is sufficiently expressive to distinguish all outerplanar graphs. Hence, any GNN that matches the expressivity of 3-WL, such as 3-IGN (Maron et al., 2019) or 3-GNN (Morris et al., 2019), is capable of solving our main goal of distinguishing all outerplanar graphs. However, a single iteration of a naive implementation of the 3-WL algorithm on a graph with $n$ vertices is at least $\mathcal{O}(n^3)$ (Immerman & Lander, 1990; Kiefer, 2020), which can be infeasible even for medium-sized real-world graphs. Similarly, a single 3-GNN or 3-IGN layer runs in roughly $\mathcal{O}(n^3)$ time (Maron et al., 2019; Morris et al., 2019). Even when additionally restricting the graph class to *biconnected* outerplanar graphs, MPNNs are not sufficiently expressive. Furthermore, Zhang et al. (2023b) have shown that most GNNs cannot even detect simple properties associated with biconnectivity such as articulation vertices. They find that only their distance-based GNN and specific GNNs based on subgraphs (Bevilacqua et al., 2021; Frasca et al., 2022) are able to detect some of these properties. Again, these approaches have at least quadratic worst case runtime.

It is not straightforward to use outerplanarity to efficiently improve the expressivity of GNNs, as even finding a subgraph remains NP-hard for outerplanar graphs (Sysło, 1982). Thus, methods like the graph structural network (Bouritsas et al., 2022) that rely on counting subgraphs remain computationally expensive even on outerplanar graphs. Subgraph GNNs model graphs as a collection of subgraphs (Frasca et al., 2022), which usually requires a pre-processing with at least quadratic runtime, depending on the method used to extract

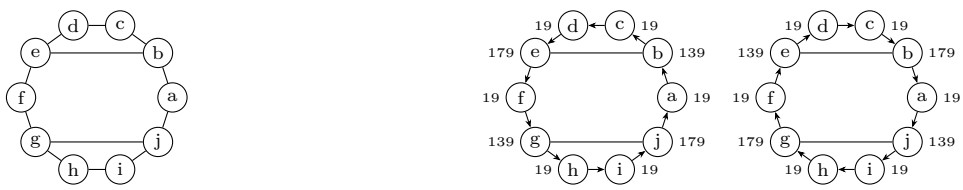

Figure 2: A graph and both directions of its directed Hamiltonian cycle. Nodes are annotated with their HALs, the distances on the Hamiltonian cycle to their neighbors (Colbourn & Booth, 1981).

subgraphs. For example, Node-delete (Bevilacqua et al., 2021) creates all subgraphs that are obtained by deleting a single node creating $\mathcal{O}(n^2)$ nodes in total. $k$-ego-net (Bevilacqua et al., 2021) extracts the $k$-hop neighborhood for each node which for $k \geq 2$ can create $\mathcal{O}(n^2)$ nodes in the worst case, for example for star graphs, which are also outerplanar.

Recently, Dimitrov et al. (2023) have proposed PlanE, a GNN architecture that can distinguish all planar graphs in $\mathcal{O}(n^2)$ time. As a consequence, PlanE is also able to distinguish all outerplanar graphs. However, this comes at the cost of (1) runtime, (2) flexibility, and—counter-intuitively—(3) expressivity. (1) PlanE requires a quadratic time pre-processing whereas our proposed CAT+MPNN runs in linear time. (2) PlanE is a GNN architecture while CAT is a graph transformation. This means, that CAT can be combined with any GNN while PlanE requires specialized GNN layers which are not easily combined with other architectures. (3) Without changes to PlanE's pre-processing, PlanE is incapable of operating on graphs with non-planar components whereas CAT still achieves at least 1-WL expressivity on such graphs. This leads to the counter-intuitive result that vanilla PlanE is incomparable in expressivity to CAT (see Prop. 2 and 3 in the Appendix).

Finally, while there exist many GNNs that are provably more expressive than WL, little is known about the precise class of graphs for which such GNNs are provably maximally expressive. Furthermore, proving an upper bound on the expressivity of an architecture is considered difficult and requires significant effort as demonstrated by Zhang et al. (2023a). In contrast, we identify outerplanar graphs as a large practical graph family that our proposed method CAT can distinguish.

## 3 Preliminaries

A *graph* $G = (V, E, \mu, \nu)$ consists of a set of nodes $V$, a set of edges $E \subseteq V \times V$ and attributes (also called features) for the nodes $\mu \colon V \to X$ and edges $\nu \colon E \to X$, respectively, where $X$ is a set of arbitrary attributes. We refer to an edge from $u$ to $v$ by $uv$ and in case of undirected graphs $uv = vu$. The *in-neighbors* of a node $u \in V$ are denoted by $N_{\text{in}}(u) = \{v \mid vu \in E\}$. The *out-neighbors* of a node $u \in V$ are denoted by $N_{\text{out}}(u) = \{v \mid uv \in E\}$ and in case of undirected graphs, $N_{\text{in}} = N_{\text{out}}$. In this paper, the input graphs are undirected and are transformed into directed ones. A graph $G' = (V', E', \mu', \nu')$ is a subgraph of a graph $G$, denoted by $G' \subseteq G$, iff $V' \subseteq V$, $E' \subseteq E$, $\forall v \in V' : \mu'(v) = \mu(v)$, and $\forall e \in E' : \nu'(e) = \nu(e)$. A (directed) cycle $(v_1, \ldots, v_k)$ is a sequence of $k \geq 3$ distinct nodes, with $\forall i \in \{1, \ldots, k-1\} : v_i v_{i+1} \in E$ and $v_k v_1 \in E$. A graph is *acyclic* if it does not contain a cycle. Given a graph $G$, we denote the shortest path distance between two nodes $u$ and $v$ by $d_G(u, v)$, or $d(u, v)$ if $G$ is clear from the context. We denote the *diameter* of a graph $G$ by $\Phi(G) = \max_{u,v \in V(G)} d(u, v)$.

A graph is *outerplanar* if it can be drawn in the plane without edge crossings and with all nodes belonging to the exterior face (see Appendix A for details). We call an undirected graph with at least three vertices *biconnected* if the removal of any single node does not disconnect the graph. A *biconnected component* is a maximal biconnected subgraph. We refer to the outerplanar biconnected components of a graph as *blocks*.

Two graphs $G$ and $H$ are isomorphic, if there exists a bijection $\psi \colon V(G) \to V(H)$, so that $\forall u, v \in V(G) \colon \mu(v) = \mu(\psi(v)) \wedge uv \in E(G) \Leftrightarrow \psi(u)\psi(v) \in E(H) \wedge \forall uv \in E(G) \colon \nu(uv) = \nu(\psi(u)\psi(v))$. We call $\psi$ an isomorphism between $G$ and $H$. An *in-tree* $T$ is a directed, acyclic graph with a distinct *root*

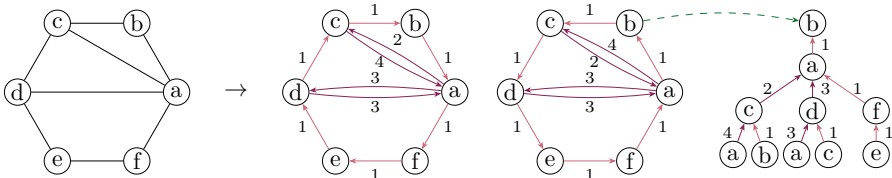

Figure 3: Biconnected outerplanar graph $G$, $\text{CAT}^*(G)$, and unfolding tree of node $b$.

having no outgoing edges and all other nodes having one outgoing edge and for every node there is exactly one path from it to the root.

**Weisfeiler-Leman.** The 1-dimensional Weisfeiler-Leman algorithm (WL) assigns colors (usually represented by numbers) to nodes. The color of a node $v \in V(G)$ is updated iteratively according to $c_{i+1}(v) = h\left(c_i(v), \{\!\{(\nu(uv), c_i(u)) \mid u \in N_{\text{in}}(v)\}\!\}\right)$, where $h$ is an injective function and $c_0 = \mu$. Note, that this variant of WL makes use of edge features and works on directed graphs. While traditionally WL is defined for undirected and unlabeled graphs, this is a common assumption in similar lines of work.

The *unfolding tree* with height $i$ of a node $v \in V(G)$ is defined as the in-tree $F_i^v = (v, \{\!\{F_{i-1}^u \mid u \in N_{\text{in}}(v)\}\!\})$, where $F_0^v = (\{v\}, \emptyset)$. The unfolding trees $F_i^v$ and $F_i^w$ of two nodes $v$ and $w$ are isomorphic if and only if the colors of the nodes in iteration $i$ are equal, see, e.g., Kriege (2022). The Weisfeiler-Leman algorithm has historically been used as a heuristic for graph isomorphism. Let $\text{WL}(G) = \{\!\{c_\infty(v) \mid v \in V(G)\}\!\}$ be the multiset of node colors in the stable coloring (Arvind et al., 2015). Two graphs $G$ and $H$ are not isomorphic if $\text{WL}(G) \neq \text{WL}(H)$. However, non-isomorphic graphs $G$ and $H$ with $\text{WL}(G) = \text{WL}(H)$ exist. WL for example cannot distinguish the molecular graphs in Figure 1 or a 6-cycle from two triangles.

**Expressivity.** Let $\mathcal{G}$ denote the set of all graphs and $\mathcal{G}_n = \{G \in \mathcal{G} \mid |V(G)| \leq n\}$ for all $n \in \mathbb{N}$. Let $\phi, \psi$ be two graph embedding algorithms, which map graphs to embedding spaces (e.g., $\mathbb{R}^d$). We say $\phi$ *is at least as expressive as* $\psi$ if for all pairs of graphs $G, G'$ with $\psi(G) \neq \psi(G')$ it holds that $\phi(G) \neq \phi(G')$. Let $\mathcal{G}' \subseteq \mathcal{G}$ be a family of graphs, for example, the set of all outerplanar graphs. We say that a graph embedding algorithm $\phi$ is *maximally expressive* for $\mathcal{G}'$ if for every non-isomorphic pair of graphs $G, G' \in \mathcal{G}'$ it holds that $\phi(G) \neq \phi(G')$. We can generalize this to parameterized graph embeddings such as GNNs: Let $\phi_w$ be a graph embedding with parameters $w$ (e.g., the weights of a neural network). A parameterized graph embedding $\phi_w$ is maximally expressive for $\mathcal{G}'$ if for all $n \in \mathbb{N}$ there exists a parameter choice $w_n$ such that for every non-isomorphic pair of graphs $G, G' \in \mathcal{G}' \cap \mathcal{G}_n$ it holds that $\phi_{w_n}(G) \neq \phi_{w_n}(G')$.

**Hamiltonian adjacency lists.** A Hamiltonian cycle of a graph is a cycle containing each node exactly once. A Hamiltonian cycle $(v_1, \ldots, v_k)$ on an undirected graph, can be separated into two directed Hamiltonian cycles $C = (v_1, \ldots, v_k)$ with corresponding edges $v_1 v_2, \ldots, v_k v_1$ and $\overleftarrow{C} = (v_k, \ldots, v_1)$ with corresponding edges $v_k v_{k-1}, \ldots, v_1 v_k$. Biconnected outerplanar graphs have a unique Hamiltonian cycle that can be found in linear time (Mitchell, 1979). Hamiltonian adjacency lists (HALs) are derived by annotating each node with the sorted distances $d_C$ to all its neighbors on the two directed variants of the Hamiltonian cycle $C$. Figure 2 shows a graph annotated with its HALs in both directions of the Hamiltonian cycle. Following the Hamiltonian cycle in one direction and concatenating the HALs gives a HAL sequence $S$ (and a reverse sequence $R$, for the other direction).

A sequence $S$ of length $n$ is a *cyclic shift* of another sequence $S'$ of length $n$ if there exists an $\ell \in \mathbb{N}$ such that $S_i = S'_j$ for all $i \in \{1, \ldots, n\}$ where $j = i + \ell \mod n$. The HAL sequence uniquely identifies a biconnected outerplanar graph (if both directions and cyclic shifts are considered):

**Lemma 1** (Colbourn & Booth 1981). *Two biconnected outerplanar graphs $G$ and $H$ with HAL and reverse sequences $S_G, S_H$ and $R_G, R_H$ are isomorphic, iff $S_G$ is a cyclic shift of $S_H$ or $R_H$.*

Table 1: Common benchmark datasets and percentage of outerplanar graphs in them.

| Dataset | #Graphs | Outerplanar |
|---|---|---|
| ZINC | 12000 | 98 % |
| PCQM-Contact | 529434 | 98 % |
| MOLESOL | 1128 | 97 % |
| MOLTOXCAST | 8576 | 96 % |
| MOLTOX21 | 7831 | 96 % |
| MOLLIPO | 4200 | 96 % |
| MOLCLINTOX | 1477 | 94 % |
| NCI-2000 | 250251 | 94 % |
| peptides-func | 15535 | 93 % |
| MOLBACE | 1513 | 93 % |
| MOLSIDER | 1427 | 92 % |
| MOLBBBP | 2039 | 92 % |
| MOLHIV | 41127 | 92 % |

Table 2: Pre-processing time of CAT on the training splits of all datasets and relative additional training/evaluation time with CAT.

| Dataset | CAT Runtime | Train+Eval. Time Change |
|---|---|---|
| MOLESOL | $2 \pm 1$ s | 26 % |
| MOLBBBP | $5 \pm 1$ s | 36 % |
| MOLSIDER | $6 \pm 1$ s | 21 % |
| MOLBACE | $6 \pm 1$ s | 42 % |
| MOLLIPO | $14 \pm 1$ s | 38 % |
| MOLTOX21 | $15 \pm 1$ s | 27 % |
| MOLTOXCAST | $16 \pm 1$ s | 13 % |
| ZINC | $44 \pm 1$ s | 27 % |
| MOLHIV | $152 \pm 1$ s | 31 % |

## 4 Identifying Outerplanar Graphs Using Weisfeiler-Leman

We develop a graph transformation called *cyclic adjacency transform* (CAT), that enables WL to distinguish all outerplanar graphs. We first introduce CAT*, enabling WL to distinguish any pair of non-isomorphic biconnected outerplanar graphs, and then extend it to all outerplanar graphs.

In CAT* (see Section 4.1), nodes are duplicated to represent the Hamiltonian cycle in both directions. We annotate edges not in the Hamiltonian cycle with the distance their endpoints have on the Hamiltonian cycle. This allows the Weisfeiler-Leman algorithm to encode HAL sequences in the unfolding trees of the nodes and in turn distinguish pairs of non-isomorphic biconnected outerplanar graphs.

To extend our transformation to all outerplanar graphs (see Section 4.2), we need to ensure that the biconnected components keep their unique encoding. It should also be ensured that non-isomorphic graphs with the same biconnected components (but arranged or rotated differently) can be distinguished. We address this by introducing articulation and block pooling vertices. These nodes contain all the information about their respective blocks and their position in these blocks. Together with nodes outside of blocks, this forms a forest which encodes the entire original graph. As it is known that WL is maximally expressive on forests, it follows that WL is maximally expressive on such graphs.

### 4.1 Identifying Biconnected Outerplanar Graphs Using Weisfeiler-Leman

We first present a graph transformation called CAT*, that allows the Weisfeiler-Leman algorithm to distinguish any two non-isomorphic *biconnected* outerplanar graphs. Figure 3 shows an example of CAT*. Note that CAT*($G$) consists of two disjoint copies of $G$, with directed and annotated edges. We first describe the transformation informally for ease of understanding, followed by the formal definition.

We start with an empty graph and add the (unique) Hamiltonian cycle of the original graph twice, directed in opposite ways (steps 1 and 2). Then, we add the remaining edges of the original graph to both cycles, directed in both directions each (step 3). Finally, we set node and edge features to the original features, and extend edge features with the distance of their endpoints in the Hamiltonian cycle (step 4).

**Definition 1.** *The* CAT*($G$) = $G'$ *transformation maps a biconnected outerplanar graph* $G = (V, E, \mu, \nu)$ *to a new graph* $G' = (V', E', \mu', \nu')$ *as follows:*

1. *Let* $(v_1, \ldots, v_n)$ *be the Hamiltonian cycle of* $G$ *and* $C$ *and* $\overleftarrow{C}$ *be its directed versions.*

2. *Add node and edge disjoint copies of* $C$ *and* $\overleftarrow{C}$ *together with the corresponding edges to an empty graph* $G'$ *and* $\forall e \in E'$ *set* $\nu'(e) = (1, \nu(e))$.

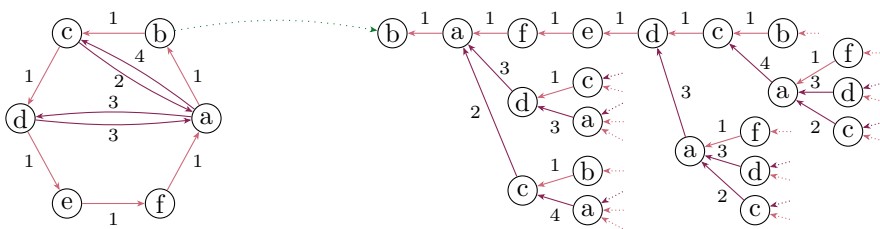

Figure 4: One part of the CAT* transformation of the graph from Figure 3 and an example unfolding tree of one of its nodes from which the HAL sequence of the original graph can be reconstructed.

3. *Let $D \subseteq E$ be the edges of $G$ not on the (undirected) Hamiltonian cycle. Add edges in both directions to $G'$ for the copies of $C$ and $\overleftarrow{C}$ for each edge in $D$: $E' = E' \cup E_d \cup E_{\overleftarrow{d}}$ with $E_d = \bigcup_{v_i v_j \in D} \{v'_i v'_j, v'_j v'_i\}$ and $E_{\overleftarrow{d}} = \bigcup_{v_i v_j \in D} \{v''_i v''_j, v''_j v''_i\}$ for copies $v'_i$ of $v_i$ in $C$ (resp. $v''_i$ in $\overleftarrow{C}$).*

4. *$\forall v'_i, v''_i \in V'$ set $\mu'(v'_i) = \mu'(v''_i) = \mu(v_i)$. $\forall v_i v_j \in E_d$ set $\nu'(v_i v_j) = (d_C(v_j, v_i), \nu(v_i v_j))$ and $\forall v'_i v'_j \in E_{\overleftarrow{d}}$ set $\nu'(v'_i v'_j) = (d_{\overleftarrow{C}}(v'_j, v'_i), \nu(v_j v_i))$.*

Using CAT* we prove our first main result.

**Theorem 1.** *Two biconnected outerplanar graphs $G$ and $H$ are isomorphic, if and only if* $\mathrm{WL}(\mathrm{CAT}^*(G)) = \mathrm{WL}(\mathrm{CAT}^*(H))$.

*Proof.* Two graphs are distinguished by WL if and only if the multisets of node colors of their stable colorings differ. Trivially, $|V(G)| \neq |V(H)| \Rightarrow |V(\mathrm{CAT}^*(G))| \neq |V(\mathrm{CAT}^*(H))| \Rightarrow \mathrm{WL}(\mathrm{CAT}^*(G)) \neq \mathrm{WL}(\mathrm{CAT}^*(H))$, so we only focus on graphs with $|V(G)| = |V(H)|$. Two nodes only get the same color if their unfolding trees are isomorphic. The first number in the HAL of each node is always 1, so it can be ignored, and the last number is always $|V(G)| - 1$, so this can simply be reconstructed by $|V(\mathrm{CAT}^*(G))|$. The rest of the HAL sequence and the node labels of $G$ can be reconstructed from the unfolding tree of any node in $\mathrm{CAT}^*(G)$: Trivially, each node has two direct neighbors in the Hamiltonian cycle. In the unfolding tree these are the parent and the single child with the 1-annotated edge. All other neighbors in the HAL can be reconstructed by looking at the weights of the edges that do not have weight 1. Figure 4 shows an example. Looking at any two biconnected outerplanar graphs with $n$ nodes, the Weisfeiler-Leman algorithm will be able to distinguish them after at most $n$ iterations, iff they are non-isomorphic: Since the HAL sequence is encoded in the unfolding trees from all starting points (cyclic shift) and in both directions (reverse direction), this identifies isomorphism by Lemma 1. □

## 4.2 Extending CAT* to All Outerplanar Graphs

While CAT* is defined for single blocks (biconnected outerplanar components), an outerplanar graph can contain multiple of them, as well as additional connections. We define the CAT transformation by applying CAT* to the blocks of the graphs and adding additional nodes and edges, which enable WL to distinguish any two non-isomorphic outerplanar graphs. We first give an informal description to provide an intuition for the various steps before stating the formal definition.

Starting from an empty graph, add the graph induced by all edges not in blocks and nodes in more than one block (steps 1 and 2). For each block, add the result of CAT* applied to the block (step 3.1). We refer to nodes created in this step as Hamiltonian cycle nodes. Add pooling nodes connecting the two Hamiltonian cycle nodes corresponding to the same original node (step 3.2). Add a node for each block (block nodes) and connect it to the pooling nodes of that block (step 3.3). Relabel the nodes from step 2, that belong to at least one block (articulation nodes) and connect them to the pooling nodes corresponding to the same original node (step 3.6). Finally, create a global (block) pooling node and connect it to the block nodes (step 4). For each node we add, its initial features (if present) are extended by a label referring to the type (Hamiltonian cycle node, pooling node, articulation node, etc.).

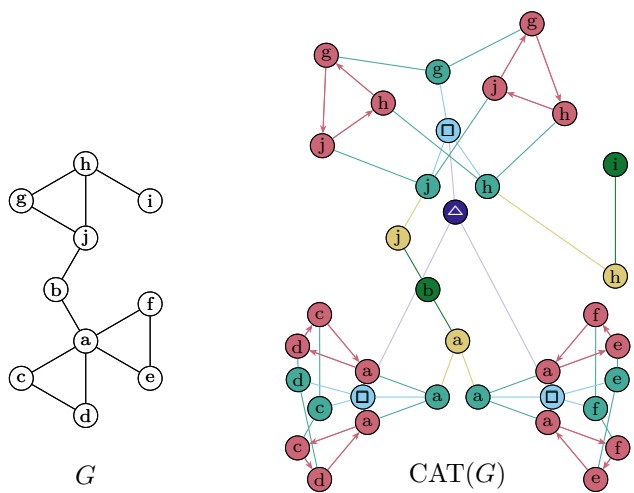

Figure 5: A graph and its CAT transformation. Original node labels are represented by letters, edge and node labels from the CAT transform are represented by colors. Note that CAT($G$) looks like a cat.

**Definition 2.** *The* CAT($G$) = $G'$ *transformation maps a graph $G$ to a new graph $G'$ as follows:*

1. *Let $B_1, \ldots, B_\ell$ be the blocks of $G$ and let $F$ be the graph induced by the edges of $G$ that are not in any block plus the nodes that are present in more than one block. Let $\{\perp, \square, \bowtie, \star, \triangle\}$ be distinct node labels not in $X$.*

2. *Add $\underline{F}$ to $G'$ with labels $\underline{\mu'(v) = (\perp, \mu(v))}$ for all $v \in F$.*

3. *For each block $B_i$ in $G$:*

   3.1. *Let $\underline{B_i', \overleftarrow{B_i'}}$ be the two connected components in CAT$^*(B_i)$. Add $\underline{B_i'$ and $\overleftarrow{B_i'}}$ to $G'$.*

   3.2. *For all pairs $(v, \overleftarrow{v})$ of corresponding nodes in $B_i'$ and $\overleftarrow{B_i'}$ add a node $p_v$ with $\underline{\mu'(p_v) = (\star, \mu(v))}$ and edges $\underline{p_v v, v p_v, p_v \overleftarrow{v}, \overleftarrow{v} p_v}$ to $G'$.*

   3.3. *Add a node $\underline{b_i}$ to $G'$ with $\mu'(b_i) = \square$. For all $v \in V(B_i)$ add edges $\underline{b_i p_v}$ and $\underline{p_v b_i}$.*

   3.4. *Let $A_i = V(B_i) \cap V(F)$ be the nodes of $B_i$ in $F$.*

   3.5. *Let $\gamma_i : A_i \to V(B_i')$ map nodes of $F$ to their copy in $B_i'$.*

   3.6. *For each $\underline{a \in A_i}$, let $\underline{\mu'(a) = (\bowtie, \mu(a))}$ and add edges $\underline{p_{\gamma_i(a)} a}$ and $\underline{a p_{\gamma_i(a)}}$ to $G'$.*

4. *Add node $\underline{g}$ with $\mu'(g) = \triangle$ to $G'$ and for all nodes $\underline{b_i}$, add edges $\underline{g b_i}$ and $\underline{b_i g}$ to $G'$.*

5. *Let* CAT($G$) = $G'$.

An example of the CAT transformation can be seen in Figure 5. Appendix F contains additional visualizations of the transformation on real-life molecular graphs. Informally, a graph is transformed using CAT by uniquely encoding all its blocks (using CAT$^*$) and then connecting the results to encode also the position within the graph and the orientation using the original graph structure. All nodes and edges that are not part of a block are of course also added to the transformed graph.

We can now prove our main result, which implies that MPNNs together with CAT are maximally expressive on outerplanar graphs.

**Theorem 2.** *Two outerplanar graphs $G$ and $H$ are isomorphic, if and only if* WL(CAT($G$)) = WL(CAT($H$)).

*Proof.* Following Theorem 1, each block will be uniquely identified by WL. Since the additional nodes have distinct labels, they will not cause WL to falsely report two blocks as isomorphic when they are not. The

Table 3: Resistance and diameter before and after the CAT transformation. $\overline{\rho(G)}$ and $\max_{\rho(G)}$ denote the average and maximum pair-wise effective resistance for graph $G$. Results are reported as mean and standard deviation across all graphs in the datasets. In all cases, smaller is better.

| Dataset | $\Phi(G)$ | $\Phi(\mathrm{CAT}(G))$ | $\overline{\rho(G)}$ | $\overline{\rho(\mathrm{CAT}(G))}$ | $\max_{\rho(G)}$ | $\max_{\rho(\mathrm{CAT}(G))}$ |
|---|---|---|---|---|---|---|
| ZINC | $12.5 \pm 2.6$ | $\mathbf{9.9 \pm 1.6}$ | $4.0 \pm 0.7$ | $\mathbf{2.6 \pm 0.4}$ | $10.0 \pm 2.0$ | $\mathbf{7.7 \pm 1.9}$ |
| MOLESOL | $\mathbf{6.6 \pm 3.3}$ | $6.9 \pm 3.8$ | $2.3 \pm 1.0$ | $\mathbf{2.1 \pm 0.9}$ | $\mathbf{5.5 \pm 2.3}$ | $6.0 \pm 2.8$ |
| MOLTOXCAST | $8.5 \pm 4.7$ | $\mathbf{8.4 \pm 4.0}$ | $3.0 \pm 1.5$ | $\mathbf{2.6 \pm 1.3}$ | $\mathbf{7.2 \pm 4.3}$ | $7.4 \pm 3.9$ |
| MOLTOX21 | $8.8 \pm 4.6$ | $\mathbf{8.7 \pm 4.0}$ | $3.1 \pm 1.5$ | $\mathbf{2.7 \pm 1.3}$ | $\mathbf{7.5 \pm 4.2}$ | $7.7 \pm 3.9$ |
| MOLLIPO | $13.8 \pm 4.0$ | $\mathbf{9.9 \pm 2.1}$ | $4.3 \pm 1.2$ | $\mathbf{2.6 \pm 0.5}$ | $10.7 \pm 3.4$ | $\mathbf{7.9 \pm 2.3}$ |
| MOLBACE | $15.1 \pm 3.2$ | $\mathbf{11.5 \pm 2.8}$ | $5.0 \pm 1.3$ | $\mathbf{2.9 \pm 0.7}$ | $12.5 \pm 3.4$ | $\mathbf{9.1 \pm 2.6}$ |
| MOLSIDER | $12.6 \pm 11.8$ | $\mathbf{10.4 \pm 7.3}$ | $4.1 \pm 3.8$ | $\mathbf{2.9 \pm 2.2}$ | $10.4 \pm 11.0$ | $\mathbf{8.9 \pm 6.8}$ |
| MOLBBBP | $10.7 \pm 3.7$ | $\mathbf{9.1 \pm 2.6}$ | $3.4 \pm 1.1$ | $\mathbf{2.4 \pm 0.6}$ | $8.3 \pm 3.8$ | $\mathbf{7.5 \pm 2.5}$ |
| MOLHIV | $11.9 \pm 5.2$ | $\mathbf{9.9 \pm 3.8}$ | $3.9 \pm 1.7$ | $\mathbf{2.7 \pm 1.2}$ | $9.3 \pm 4.7$ | $\mathbf{8.2 \pm 3.8}$ |

information about the entire HAL sequence of each block is stored in the block nodes $b$. The pooling nodes $p$ connect the block and block nodes to the rest of the graph (through the articulation nodes $a$), determining the orientation of the block. Note that the graph returned by CAT without the CAT* blocks and the global pooling node $g$ is a tree. Relying on the labels of the pooling and block nodes, we can reconstruct the original graph from this tree. As WL can distinguish non-isomorphic labeled trees (Arvind et al., 2015; Kiefer, 2020), it can thus distinguish non-isomorphic outerplanar graphs using CAT. For the other direction, note that CAT is permutation-invariant: for two isomorphic graphs $G$ and $H$, the graphs $\mathrm{CAT}(G)$ and $\mathrm{CAT}(H)$ are isomorphic and WL will give the same coloring for both. □

Importantly, we can compute $\mathrm{CAT}(G)$ in linear time. The computational complexity is dominated by the computation of the blocks (Tarjan, 1972) and their Hamiltonian cycles (Mitchell, 1979), which both require linear time. Note that we only add a linear number of nodes and edges. From Morris et al. (2019) and Xu et al. (2019) it follows, that MPNNs, that are as expressive as 1-WL, can distinguish $\mathrm{CAT}(G)$ and $\mathrm{CAT}(H)$ for non-isomorphic outerplanar graphs $G$ and $H$.

Note that our proof used an important property of the WL algorithm: Adding uniquely labeled nodes and edges to WL-distinguishable graphs never leads to WL-indistinguishable graphs. We use this property to add a global pooling node in step 4 of CAT which is connected to all block pooling nodes. This allows to pass messages between block nodes in fewer iterations in the subsequent MPNN step.

CAT can also be applied to non-outerplanar graphs. In this case, our graph transformation performs the steps described in Definition 2. However, if a non-outerplanar biconnected component $B_i$ is encountered, only one copy $B_i'$ is created in $\mathrm{CAT}(G)$ and its vertices are connected to the corresponding pooling nodes. An example for this is depicted in Appendix D. While this never reduces expressivity, it is also not guaranteed to improve expressivity on non-outerplanar graphs. Note that it can be determined in linear time whether a block is outerplanar while trying to compute the Hamiltonian cycle of the block (Mitchell, 1979). Hence, the CAT transformation always only requires linear time.

### 4.3 Influence of CAT on Graph Connectivity

As poor graph connectivity may negatively influence the predictive performance of GNNs (Alon & Yahav, 2021), we investigate the effects of CAT on different measures of graph connectivity such as the diameter $\Phi(G)$ of a graph. We refer to the shortest path distance between two nodes as the *distance* between them.

**Observation 1.** *For a block $B$ of a graph $G$, it holds that $\Phi(\mathrm{CAT}(B)) \leq 4$.*

*Proof.* Let $u, v \in V(\mathrm{CAT}(B))$. By definition all nodes in $\mathrm{CAT}(B)$ are either from a Hamiltonian cycle created by CAT*, a pooling node, or a block node. If both nodes are from a Hamiltonian cycle, then there is a path $u, p_u, b, p_v, v$ between them, where $p_u, p_v$ are pooling nodes and $b$ is the block node. Hence, $d_{\mathrm{CAT}(G)}(u, v) \leq 4$. If $u$ or $v$ is a pooling or a block node, then the above path implies that $d_{\mathrm{CAT}(G)}(u, v) < 4$. □

Table 4: Predictive performance of MPNNs with and without CAT on different molecular benchmark datasets. Arrows indicate whether smaller (↓) or bigger (↑) results are better. **Bold** entries are an MPNN with CAT that outperforms the same MPNN without CAT.

| Dataset →
↓ Model | ZINC
MAE ↓ | ZINC250k
MAE ↓ | MOLHIV
ROC-AUC ↑ | MOLBACE
ROC-AUC ↑ | MOLBBBP
ROC-AUC ↑ |
|---|---|---|---|---|---|
| GIN | $0.168 \pm 0.007$ | $0.033 \pm 0.003$ | $77.9 \pm 1.0$ | $74.6 \pm 3.2$ | $66.0 \pm 2.1$ |
| CAT+GIN | $\mathbf{0.101 \pm 0.004}$ | $0.034 \pm 0.003$ | $76.7 \pm 1.8$ | $\mathbf{79.5 \pm 2.5}$ | $\mathbf{67.2 \pm 1.8}$ |
| GCN | $0.184 \pm 0.013$ | $0.067 \pm 0.005$ | $76.7 \pm 1.4$ | $77.9 \pm 1.7$ | $66.1 \pm 2.4$ |
| CAT+GCN | $\mathbf{0.123 \pm 0.008}$ | $\mathbf{0.034 \pm 0.003}$ | $\mathbf{77.1 \pm 1.6}$ | $\mathbf{79.2 \pm 1.5}$ | $\mathbf{68.3 \pm 1.7}$ |
| GAT | $0.375 \pm 0.013$ | $0.103 \pm 0.004$ | $76.6 \pm 2.0$ | $81.7 \pm 2.3$ | $66.2 \pm 1.4$ |
| CAT+GAT | $\mathbf{0.201 \pm 0.022}$ | $\mathbf{0.046 \pm 0.004}$ | $75.3 \pm 1.6$ | $79.3 \pm 1.6$ | $66.0 \pm 1.9$ |

| Dataset →
↓ Model | MOLSIDER
ROC-AUC ↑ | MOLESOL
RMSE ↓ | MOLTOXCAST
ROC-AUC ↑ | MOLLIPO
RMSE ↓ | MOLTOX21
ROC-AUC ↑ |
|---|---|---|---|---|---|
| GIN | $56.6 \pm 1.0$ | $1.105 \pm 0.077$ | $65.3 \pm 0.6$ | $0.717 \pm 0.016$ | $75.8 \pm 0.7$ |
| CAT+GIN | $\mathbf{58.2 \pm 0.9}$ | $\mathbf{0.985 \pm 0.055}$ | $\mathbf{65.6 \pm 0.5}$ | $0.798 \pm 0.031$ | $74.8 \pm 1.0$ |
| GCN | $56.7 \pm 1.5$ | $1.053 \pm 0.087$ | $64.4 \pm 0.4$ | $0.748 \pm 0.018$ | $76.4 \pm 0.3$ |
| CAT+GCN | $\mathbf{57.9 \pm 1.8}$ | $\mathbf{1.006 \pm 0.036}$ | $\mathbf{66.2 \pm 0.8}$ | $0.771 \pm 0.023$ | $74.9 \pm 0.8$ |
| GAT | $58.4 \pm 1.0$ | $1.037 \pm 0.063$ | $63.8 \pm 0.8$ | $0.728 \pm 0.024$ | $76.3 \pm 0.6$ |
| CAT+GAT | $58.3 \pm 1.3$ | $1.09 \pm 0.048$ | $\mathbf{64.5 \pm 0.8}$ | $0.754 \pm 0.021$ | $75.4 \pm 0.7$ |

**Observation 2.** *Let $B_i$ and $B_j$ be two blocks of a graph $G$. In $\mathrm{CAT}(G)$, the maximum distance between any node in $\mathrm{CAT}(B_i)$ and any node in $\mathrm{CAT}(B_j)$ is 6.*

*Proof.* Let $u \in V(\mathrm{CAT}(B_i))$ and $v \in V(\mathrm{CAT}(B_j))$. If $B_i = B_j$, then Observation 1 implies $d_{\mathrm{CAT}(G)}(u, v) \leq 4$. If $B_i \neq B_j$, then there exists a path $u, p_u, b_i, g, b_j, p_v, v$ where $p_u, p_v$ are pooling nodes, $b_i, b_j$ are the block node for block $B_i, B_j$, and $g$ is the global block pooling node. Thus, $d_{\mathrm{CAT}(G)}(u, v) \leq 6$. □

Observations 1 and 2 show that for some subgraphs CAT provides constant upper bounds in diameter. On the graph level, CAT can only lead to a small increase of the diameter.

**Proposition 1.** *For an outerplanar graph $G$, $\Phi(\mathrm{CAT}(G)) \leq \Phi(G) + 7$.*

*Proof sketch.* To prove the proposition we bound the distance between any pair of nodes in $\mathrm{CAT}(G)$ by case analysis based on the type of nodes. We defer the full proof to Appendix B. □

In most practical cases, the short-cutting inside or between blocks leads to CAT reducing the graph diameter (see Observations 1 and 2). In Table 3, we demonstrate this on molecular benchmark datasets. Besides the diameter, another useful graph connectivity measure is the *effective resistance*. The notion of effective resistance originates in electrical engineering (Kirchhoff, 1847) and has implications on several graph properties. For example, the effective resistance between two nodes is proportional to the commute time between them (Chandra et al., 1989). Intuitively, a large effective resistance between two nodes suggests that information propagation between the nodes is hindered. Recently, effective resistance has been in fact linked to *over-squashing* (Black et al., 2023) in GNNs, which is a negative effect that leads to long-range interactions having little impact on the predictions of a GNN. Effective resistance as introduced by Kirchhoff (1847) is naturally only defined for undirected graphs. As CAT produces directed graphs, we therefore use an extension of effective resistance introduced by Young et al. (2015) that is applicable to directed graphs. We refer to Young et al. (2015) for more details. In Table 3 we demonstrate that CAT reduces the pair-wise effective resistance on molecular benchmark datasets.

## 5 Experimental Evaluation

We investigate whether our proposed method CAT can improve the predictive performance of MPNNs on molecular benchmark datasets.[1] We utilize three commonly used MPNNs: GIN (Xu et al., 2019), GCN (Kipf & Welling, 2017), and GAT-v2 (Veličković et al., 2018; Brody et al., 2022). We train on the commonly used ZINC (Gómez-Bombarelli et al., 2018; Sterling & Irwin, 2015) and MOLHIV (Hu et al., 2020) datasets, which contain graphs representing molecules. We supplement these with 7 small datasets (see Table 4) from the OGB collection (Hu et al., 2020). In addition to the ZINC dataset with 12k graphs we also use the larger ZINC250k variant with 250k graphs. In total, we train three MPNNs on ten datasets with and without CAT. For each configuration, we separately tune hyperparameters on the validation set and train a model with the best hyperparameters ten times on the training set and evaluate it on the test set. The only exception to this is ZINC250k where we evaluate the final hyperparameter configuration five times, due to the large dataset size. For each dataset we report the mean and standard deviation of the most common evaluation metric on the test set in the epoch with the best validation performance. For ZINC we use a batch size of 128 and an initial learning rate of $10^{-3}$ that gets halved if the validation metric does not improve for 20 epochs. The training stops after 500 epochs or if the learning rate dips below $10^{-5}$. For all other datasets we train with a fixed learning rate for 100 epochs and a batch size of 128. We use the same hyperparameter grid for all models and provide more details in Appendix E. Besides measuring the predictive performance, we also measure the time needed for applying CAT (averaged over ten runs), and the training and evaluation time for GIN and GIN+CAT with the same hyperparameters on all datasets (averaged over five runs). Finally, we report the values for the diameters and effective resistances as described in Section 4.3.

**Results.** Table 2 shows the pre-processing time of CAT. Note that this is the performance of running CAT on only a single CPU core. It is possible to achieve faster runtimes by simply parallelizing different graphs over different cores. This negligible runtime allows applying the transformation even in realistic high-throughput screening applications (Krasoulis et al., 2022), for example, on MOLHIV it takes only 5ms to process a graph. Training and prediction time on CAT transformed graphs increases by 29% on average. Table 4 shows the predictive performance of all models. Note that our baseline models obtain strong results, often surpassing the performance of (higher-order) GNNs reported in the literature, and that we train each MPNN and MPNN+CAT with exactly the same sets of hyperparameters. Overall, CAT improves the predictive performance of GIN and GCN in the majority of datasets (6 / 10 and 8 / 10, respectively). For GIN and GCN, performance increases reliably on all datasets, except MOLLIPO and MOLTOX21. Surprisingly, CAT does not work well with GAT and only improves its performance in 2 / 10 datasets. Most notably on ZINC, CAT achieves very strong results boosting the predictive performance of MPNNs between 33% (GCN) and 46% (GAT). This is only surpassed by higher-order GNNs such as CW Networks (Bodnar et al., 2021a) which obtain a MAE of $0.079 \pm 0.006$ at the cost of potentially exponential pre-processing runtime due to enumerating cycles in the graph. Table 3 shows that CAT reduces both graph diameter and maximum pair-wise resistance on most datasets. Furthermore, CAT reduces the average pair-wise resistance in all datasets. This suggests that CAT is effective at improving graph connectivity in real-life molecular graphs.

## 6 Conclusion

We proposed *Cyclic Adjacency Transform* (CAT), a linear time graph transformation, that enables the Weisfeiler-Leman algorithm to be maximally expressive on outerplanar graphs. We rely on the fact that biconnected outerplanar graphs can be uniquely identified by their Hamiltonian adjacency list sequences, which CAT encodes in unfolding trees. It follows that a combination of MPNNs and CAT can distinguish all outerplanar graphs. We achieved promising empirical results on standard molecular benchmark datasets where CAT improved the performance of GIN and GCN in most cases, while for GAT we could not observe this benefit. Computing CAT takes linear time and our implementation of CAT typically runs in the order of seconds. Motivated by the recent interest in the over-squashing phenomenon, we also studied the effect of CAT on graph connectivity. We proved that in the worst-case CAT increases the diameter of outerplanar graphs by a small additive constant. However, inspecting CAT on real-world data, we find that

---

[1] Our code can be found at https://github.com/ocatias/outerplanarGNN.

the diameter decreases most of the time. Similarly, we observed that the maximum and average pair-wise effective resistance, which is associated with over-squashing, typically decreases after applying CAT.

Our work highlights the value of GNNs designed for specific graph classes. This perspective aligns with a recent position paper, in which Morris et al. (2024) propose to investigate the runtime complexity of maximally expressive GNNs on practically relevant graphs (Challenge II.4). In general, achieving high expressivity is computationally expensive. However, for some graph classes this might not be the case. In this work, we have demonstrated that for *outerplanar* graphs maximal expressivity can be achieved in linear time.

Focusing on specific graph classes allows making use of existing graph theoretical results. This could lead to novel GNNs that are both runtime efficient and highly expressive. Possible applications are infrastructure and road networks, which are characterized by *near-planar* graphs that tend to have low *graph degeneracy* (Eppstein & Gupta, 2017). More generally, it seems promising to study sparse graphs, such as *bounded expansion* graphs (Nešetřil & De Mendez, 2012). Finally, in the future we plan to extend our work to $k$-outerplanar graphs, which are known to capture even more molecular graphs (Horváth et al., 2010).

## Acknowledgement

This work was supported in part by the Vienna Science and Technology Fund (WWTF) through projects [10.47379/VRG19009] and [10.47379/ICT22059]; by the TU Wien DK SecInt; and by the Austrian Science Fund (FWF) through project NanOX-ML (6728). MT acknowledges support from a DOC fellowship of the Austrian academy of sciences (ÖAW).

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

## A  Outerplanar Graphs

Let $G = (V, E)$ be an undirected graph. A *planar embedding* of $G$ consists of an injective mapping $f$ of nodes $V$ to vectors in the plane $\mathbb{R}^2$ and for each pair of adjacent nodes $u, v$ a continuous path between $f(u)$ and $f(v)$ such that no pair of such paths cross. More formally, a continuous path from $f(u)$ to $f(v)$ is a continuous function $p : [0, 1] \to \mathbb{R}^2$ such that $p(0) = f(u)$ and $p(1) = f(v)$. Two such continuous paths $p, p'$ cross if there exist $t, t' \in (0, 1)$ such that $p(t) = p'(t')$. The graph $G$ is *planar* if it has a planar embedding. By Fáry's theorem each planar graph always has a planar embedding with straight line segments as paths (Fáry, 1948). Intuitively a planar embedding dissects the plane into regions called *faces*. Formally these are the (topologically) connected regions of $\mathbb{R}^2$ with all the edge paths $p$ removed. Any planar embedding has a unique face not bounded by paths (the only non-compact face), which is called the *outer face*. A graph is *outerplanar* if it has a planar embedding with all nodes lying on the boundary of the outer face. See, for example, Felsner (2012) for more details.

We can also characterize outerplanar graphs using forbidden *graph minors*. A graph $H$ is a minor of $G$, if $H$ can be obtained from $G$ by a series of node deletion, edge deletion, and edge contractions (i.e., removing an edge and replacing the two endpoints by a new node). Let $K_{a,b}$ be the *complete bipartite graph* with bi-partition $A \cup B = V$ with $a = |A|$ and $b = |B|$ and $K_c$ be the *complete* graph on $c$ nodes. A graph $G$ is outerplanar if and only if $G$ has no $K_{2,3}$ minor and no $K_4$ minor. Similarly, planar graphs can be characterized by the Kuratowski theorem as graphs with no $K_{3,3}$ and no $K_5$ minor. For more details see, for example Diestel (2024).

## B  Proof of Proposition 1

We prove Proposition 1 which states that $\Phi(\mathrm{CAT}(G)) \leq \Phi(G) + 7$ for every outerplanar graph $G$.

*Proof.* Let $u, v \in V(\mathrm{CAT}(G))$ such that $d_{\mathrm{CAT}(G)}(u, v) = \Phi(\mathrm{CAT}(G))$. We call a node a *tree node* if it was not part of a block in $G$ and was not created by CAT. A node that is *not* a tree node is either a Hamiltonian cycle node, a pooling node, a block pooling node, or a global block pooling node.

**Case 1:** Both $u, v$ are not tree nodes. By Observation 2: $d_{\mathrm{CAT}(G)}(u, v) \leq 6$.

**Case 2:** Node $u$ is a tree node and $v$ is not. Let $a \in V(G)$ be the closest articulation node to $u$ in $\mathrm{CAT}(G)$. Then, there is a path of length $d_G(u, a)$ in $\mathrm{CAT}(G)$ from $u$ to $a$. We can extend this path by one node to reach a pooling node. By Definition 2, there exists a path of length at most 6 from this pooling node to $v$. Thus $d_{\mathrm{CAT}(G)}(u, v) \leq d_G(u, a) + 7 \leq \Phi(G) + 7$.

**Case 3:** Both $u, v$ are tree nodes.

**Case 3a:** Suppose that the shortest path between $u$ and $v$ in $G$ does not contain any edge inside of an outerplanar block, then $d_{\mathrm{CAT}(G)}(u, v) = d_G(u, v) \leq \Phi(G)$.

**Case 3b:** Suppose that the shortest path between $u$ and $v$ in $G$ contains one or more edges inside exactly one block. Then, we can enter and exit this block in $\mathrm{CAT}(G)$ through a path $a_1, p_1, b, p_2, a_2$, where $a_1, a_2$ are articulation nodes, $p_1, p_2$ are pooling nodes, and $b$ is a block node. Note that the articulation nodes were part of the path in $G$ which implies $d_{\mathrm{CAT}(G)}(u, v) = d_G(u, a_1) + d_G(a_2, v) + 4 = d_G(u, v) - d_G(a_1, a_2) + 4$. Furthermore, we do not need to take the one or more edges inside the block to go from $a_1$ to $a_2$. Using $d_G(a_1, a_2) \geq 1$ we obtain $d_{\mathrm{CAT}(G)}(u, v) = d_G(u, v) - d_G(a_1, a_2) + 4 \leq d_G(u, v) + 3 \leq \Phi(G) + 3$.

**Case 3c:** Suppose that the shortest path between $u$ and $v$ in $G$ contains two or more edges that are contained in two or more different blocks. Then, for $\mathrm{CAT}(G)$ we can shortcut from the first to the last block node of the shortest path between $u$ and $v$ through a path $a_1, p_1, b_1, g, b_2, p_2, a_2$, where $a_1, a_2$ are articulation nodes, $p_1, p_2$ are pooling nodes, $b_1, b_2$ are block nodes, and $g$ is the global block pooling node. Note that the articulation nodes were part of the path in $G$ which implies that $d_{\mathrm{CAT}(G)}(u, v) = d_G(u, a_1) + d_G(a_2, v) + 6 = d_G(u, v) - d_G(a_1, a_2) + 6$. By assumption, we know that $d_G(a_1, a_2) \geq 2$ which implies $d_{\mathrm{CAT}(G)}(u, v) = d_G(u, v) - d_G(a_1, a_2) + 6 \leq d_G(u, v) + 4 \leq \Phi(G) + 4$. $\qquad\square$

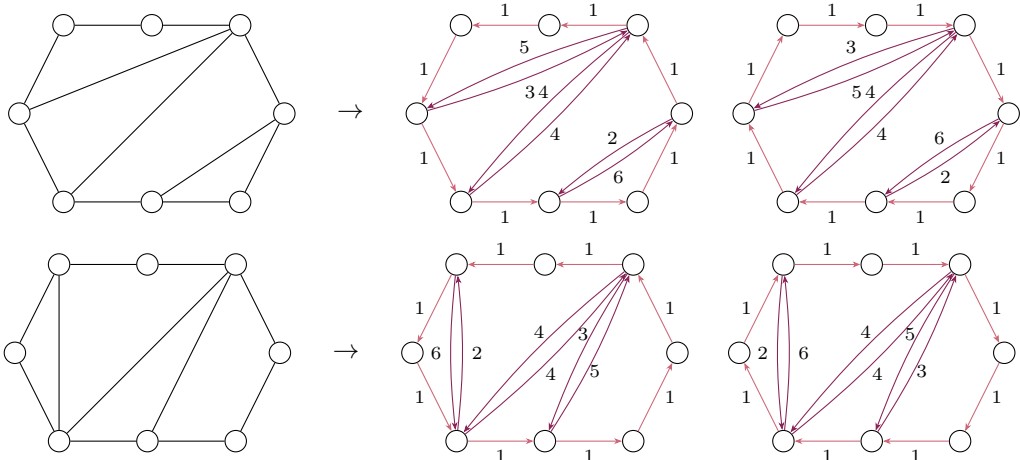

Figure 6: Additional examples for CAT*. The biconnected outerplanar graphs on the left are transformed using Definition 1.

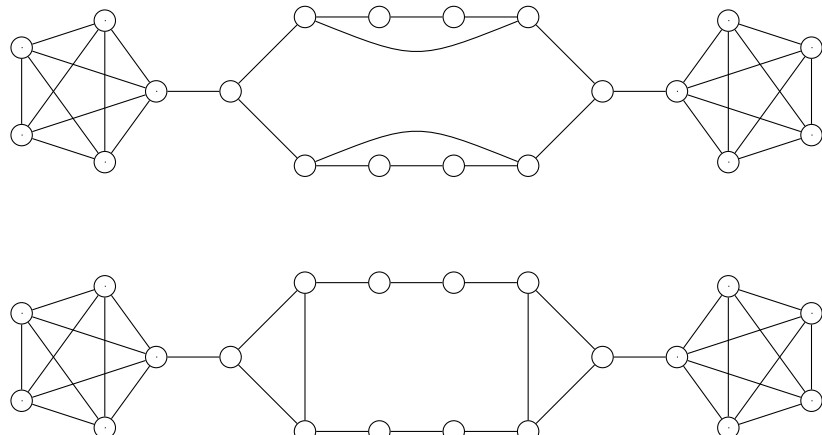

Figure 7: Two nonplanar graphs which PlanE cannot distinguish. However, CAT can distinguish the graphs by their two central outerplanar biconnected components.

## C   Additional Examples for CAT*

Here, we provide additional examples for CAT*. Figure 6 shows two graphs and the results, when transformed using CAT*. As described in Definition 1, the original graph is copied twice and the edges of the Hamiltonian cycle are directed in each direction once (with their label extended with 1). The remaining edges are added to each copy directed in both directions, with their label being extended to describe the distance of their two nodes when only using edges from the Hamiltonian cycle.

## D   Incomparable Expressivity of CAT and PlanE

The expressivity of CAT and PlanE is incomparable. This follows from Propositions 2 and 3, showing that there are pairs of graphs that are distinguishable by CAT and not PlanE and vice versa.

**Proposition 2.** *There exists non-planar graphs that* CAT *can distinguish and PlanE cannot distinguish.*

*Proof.* The claim follows from Figure 7. □

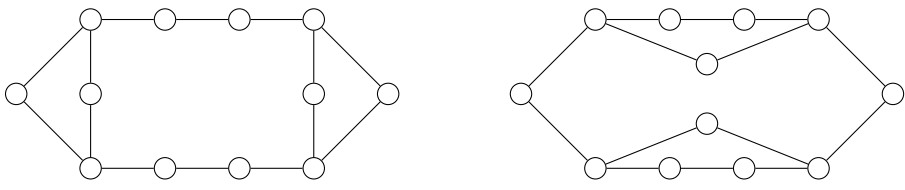

Figure 8: Two planar (but not outerplanar) graphs which CAT cannot distinguish.

**Proposition 3.** *There exists planar graphs that PlanE can distinguish and* CAT *cannot distinguish.*

*Proof.* The claim follows from Figure 8. □

## E   Details for Experimental Evaluation

Our models are implemented in PyTorch-Geometric (Fey & Lenssen, 2019) and trained on a single NVIDIA GeForce RTX 3080 GPU. We use WandB (Biewald, 2020) for tracking. The used server has 64 GB of RAM and an 11th Gen Intel(R) Core(TM) i9-11900KF CPU running at 3.50GHz. Table 5 shows the hyperparameters of our MPNNs for different datasets. We use the same hyperparameter grid for MPNNs combined with CAT. We used a smaller hyperparameter grid for MOLHIV as MOLHIV is larger than the most of the other datasets, meaning that training takes much longer. When benchmarking the speed of GIN against GIN+CAT we train for 100 epochs with a batch size of 128 on all datasets with the same hyperparameters for both models (see Table 5).

CAT **implementation.**   CAT adds an additional feature to each node which encodes the type of that node, i.e., nodes from Hamiltonian cycles, block nodes, pooling nodes, articulation nodes and or global block nodes. Furthermore, we create additional edge features encoding the types of nodes incident to this edge i.e., an edge between two different nodes in a Hamiltonian cycle has a different type than an edge from a pooling node to the block node. For newly created nodes and edges we set their remaining features to the feature of the node / edge they are based on; for example, a pooling node will have the features of the node they are performing the pooling operation for. For nodes that have no natural representation in the graph (block and block pooling nodes) we set these features to 0. To ensure that only these nodes get assigned 0 features, we shift the values of these features for all other nodes by 1.[2] Note that our MPNNs treat the distance on edges in blocks as a categorical feature. Representing the distances as numerical features did not improve performance in preliminary experiments.

Table 5: Hyperparameter grids for GIN, GCN and GAT on different datasets.

| Parameter | All datasets except MOLHIV | MOLHIV | Speed Benchmarks (Table 2) |
| --- | --- | --- | --- |
| Message passing layers | 2, 3, 4, 5 | 4, 5 | 4 |
| Final MLP layers | 2 | 2 | 2 |
| Pooling operation | mean, sum | mean, sum | mean |
| Embedding dimension | 64, 128, 256 | 64,128 | 64 |
| Jumping knowledge | last | concat | concat |
| Dropout rate | 0, 0.5 | 0.5 | 0 |

**Dataset description.**   All graphs in our datasets represent molecules and have graph-level tasks. For all tasks we use the metrics that are commonly used by the community, e.g., MAE for ZINC or ROC-AUC

---

[2]This assumes that the features are categorical and not numerical–which is the case for the used datasets.

for MOLHIV. For ZINC (10k graphs) and ZINC250k (250k graphs) (Gómez-Bombarelli et al., 2018; Sterling & Irwin, 2015) the task is to predict the constrained solubility which is a regression task. All other datasets are from the Open Graph Benchmark (Hu et al., 2020) based on MoleculeNet (Wu et al., 2018) and the task is always to predict some molecular property. On MOLHIV, the task is to predict whether a molecule inhibits HIV virus replication or not (Hu et al., 2020). For the tasks of the other datasets, we refer to Wu et al. (2018).

## F    Additional Figures

We provide additional visualizations of the CAT transformation. In all figures, the color of the vertices in the transformed graph have the following meaning: red nodes are from Hamiltonian cycles, blue nodes correspond to blocks, yellow nodes pool the nodes from Hamiltonian cycles, orange nodes correspond to articulation nodes and the gray node pools block nodes. Figure 9 shows a synthetic example with a non-outerplanar graph. Figure 10 demonstrates CAT on various synthetic graphs. Figure 11 shows the result of CAT on molecular graphs from ZINC and MOLHIV. Somewhat ironically, CAT often generates frog graphs on MOLHIV as can be seen in Figure 12.

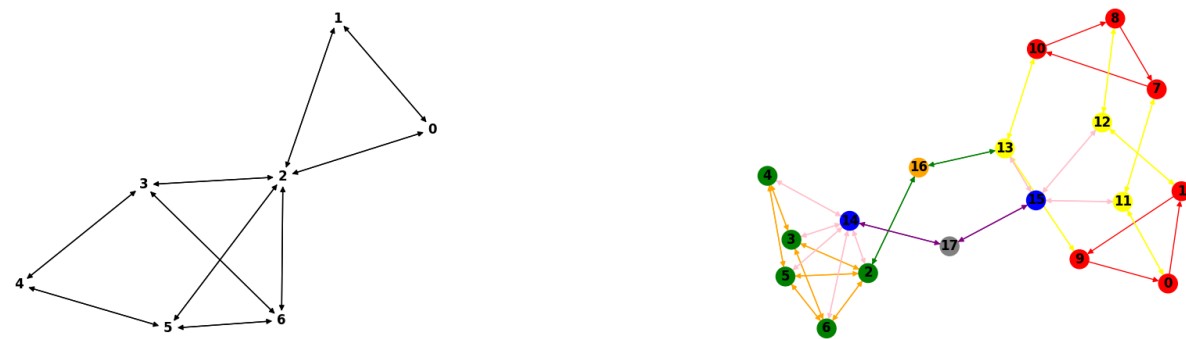

Figure 9: Left: example non-outerplanar graph. Right: result of applying CAT to the graph. Colors indicate the type of node (see Appendix F).

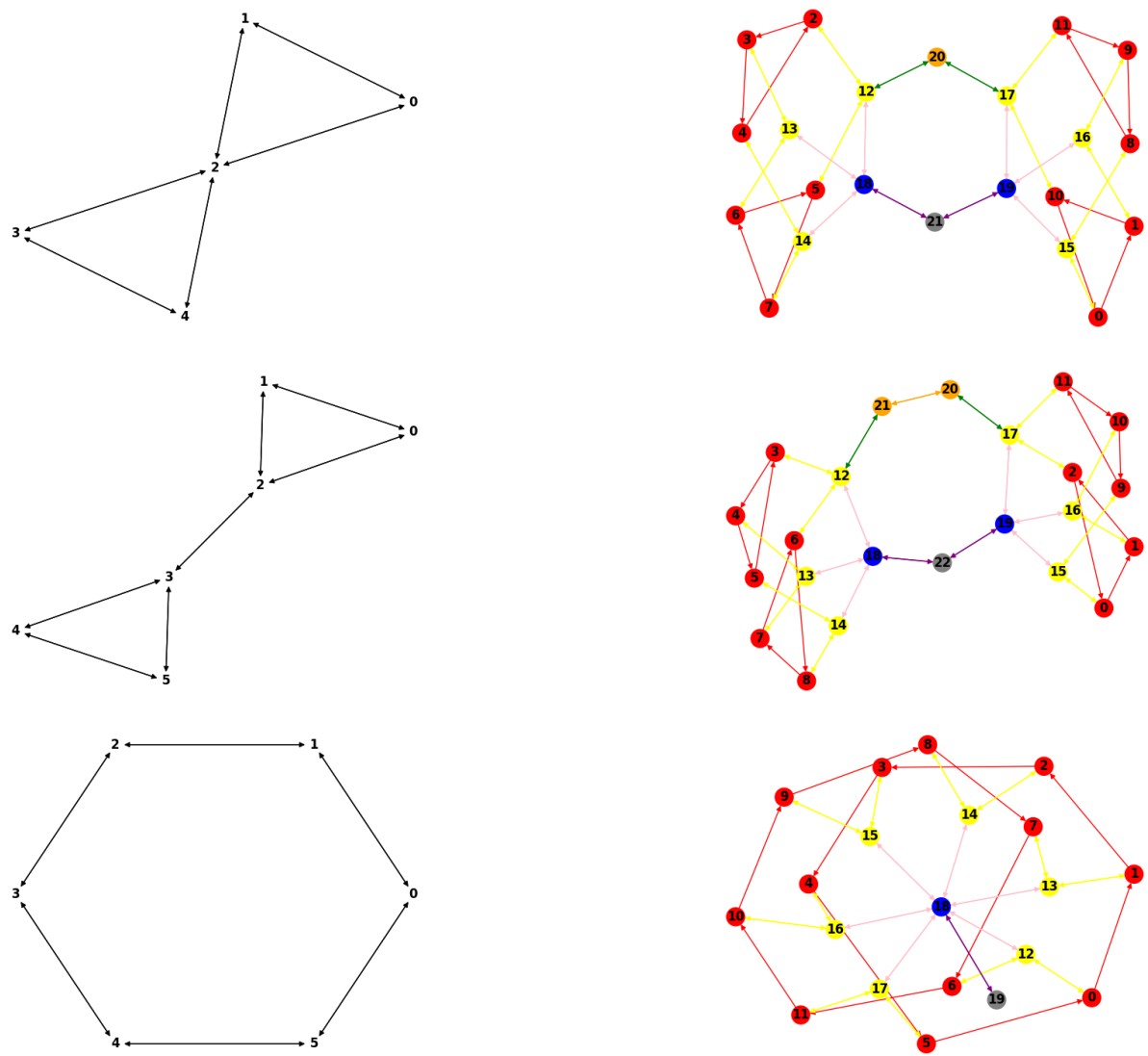

Figure 10: Left: example graphs. Right: result of applying CAT to these graphs. Colors indicate the type of node (see Appendix F).

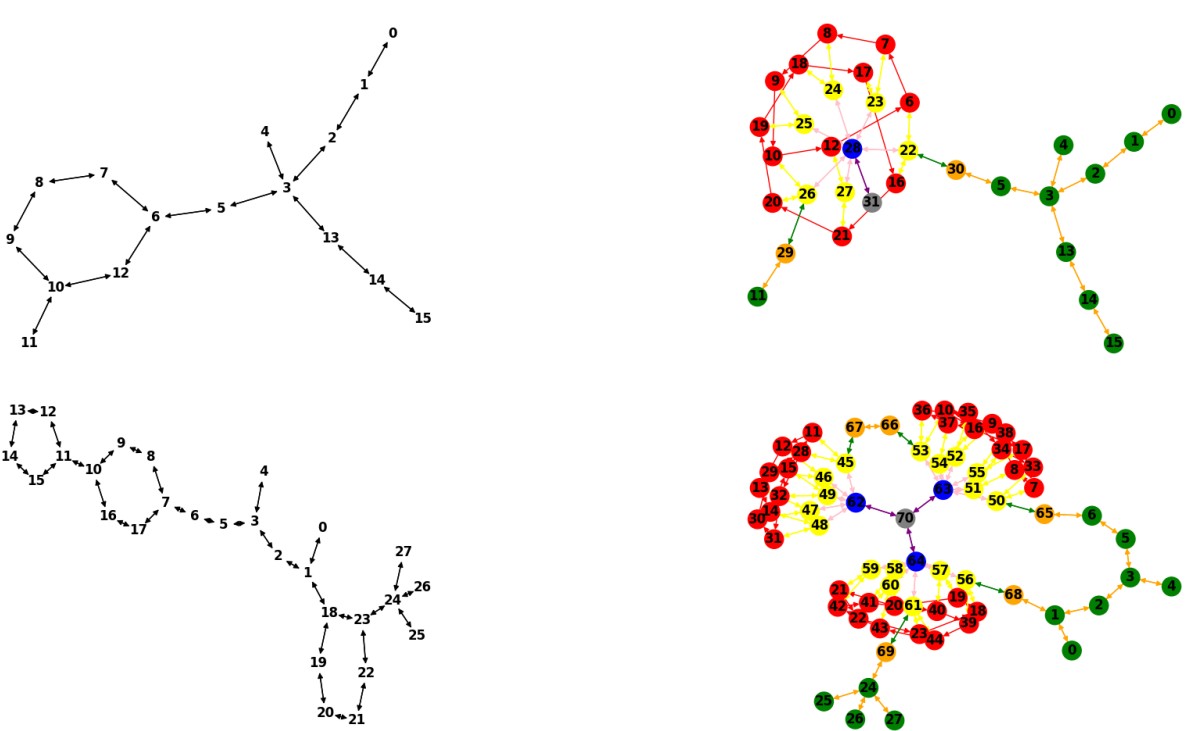

Figure 11: Left: example graphs from MOLHIV (top) and ZINC (bottom). Right: result of applying CAT to these graphs. Colors indicate the type of node (see Appendix F).

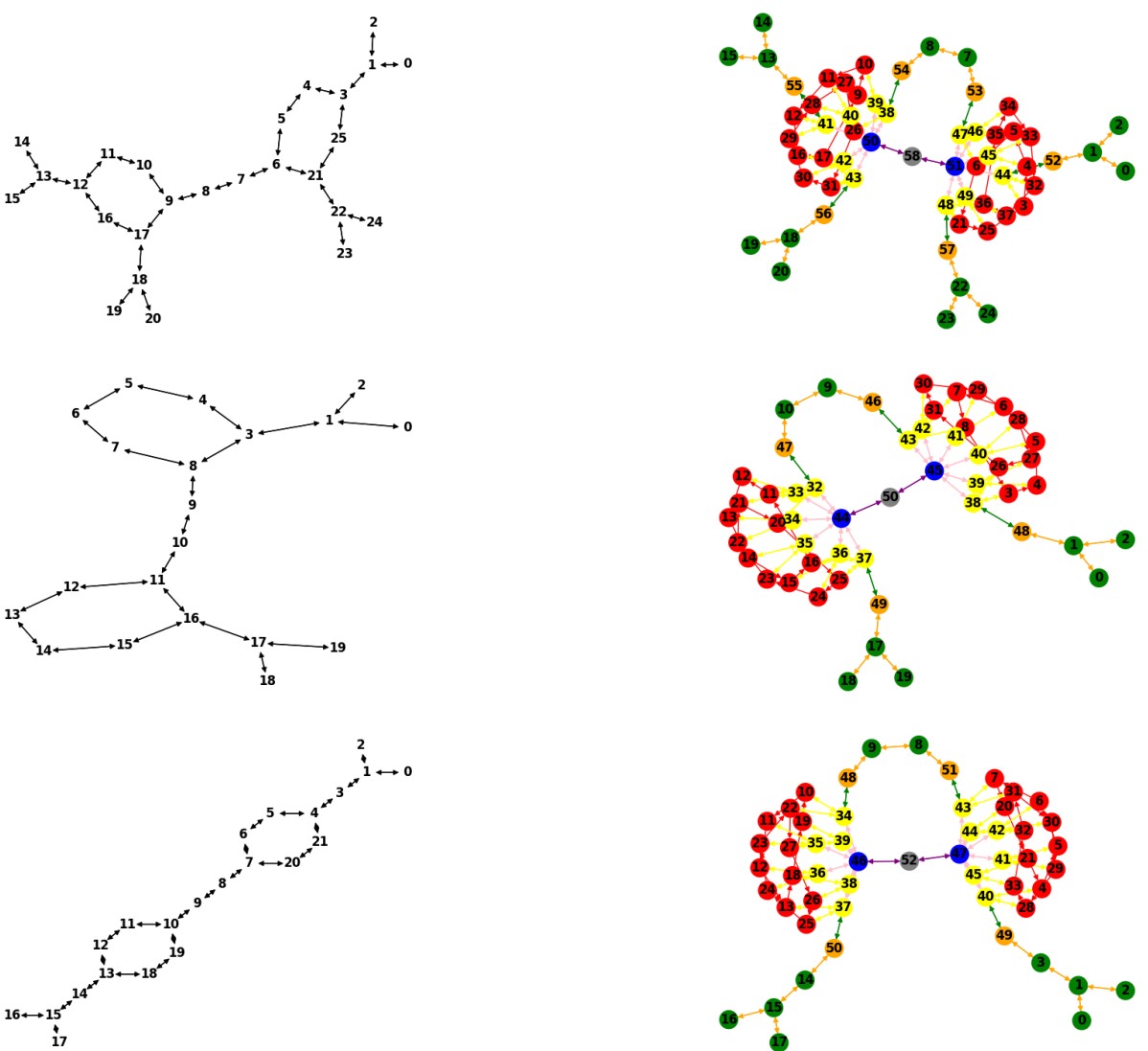

Figure 12: Left: example graphs from MOLHIV. Right: result of applying CAT to the graph. Colors indicate the type of node (see Appendix F).

