# OpenReview forum: "Maximally Expressive GNNs for Outerplanar Graphs"
_TMLR — Accepted by TMLR_

### Review · Reviewer_qtTB · 2024-09-04

**Summary Of Contributions:**

This paper introduces a transformation algorithm on graphs, called the cyclic adjacency transform. Using this transformation algorithm, this paper shows that the transformed graphs can be used along with the Weisfeiler-Leman GI test to differentiate between two isomorphic graphs.
- A graph is called outerplanar if it can be drawn in the plane without edge crossings and with all nodes belonging to the exterior face.

This cyclic adjacency transform is first applied to biconnected outerplanar graphs, then designed for all outerplanar graphs:
- A graph is called biconnected if removing any node of the graph will not cause the graph to be disconnected.
- For a biconnected outerplanar graph, the transform first obtains the directed Hamiltonian cycle of the graph and the reverse of the graph.
- In the newly transformed graph, it adds two disjoint copies of both Hamiltonian cycles.
- Then, it joins two edges belonging to the same edge in the original graph from both directions of the Hamiltonian.
- Finally, it sets the edge/node embeddings from the original graph.
- For general outerplanar graphs, the transform separates the graph into multiple blocks; then, it applies a transform similar to the above to each block. Finally, it concatenates all the transformed blocks together into the final output.

The main result shows that with this transform, the two outerplanar graphs are isomorphic if and only if their WL colorings are the same.

The paper ends with some empirical evaluations to show that this transform can improve the predictive performance of several GNNs, including GIN, GCN, and GAT.

**Audience:**

Yes

**Broader Impact Concerns:**

The focus of this manuscript is on the expressivity of GNNs for outerplanar graphs. There are no immediate implications to ethical aspects from this work (either positive or negative).

**Claims And Evidence:**

No

**Requested Changes:**

Below please find a list of editorial changes that can enhance the readability of the manuscript:

- I'm not quite sure what exactly is the problem statement/scope of this paper: My current understanding based on the main theoretical result + empirical evaluation is that this paper is designing transformation procedures that can be used to enhance the predictive performance of GNNs. Please correct me if I am wrong. If so, I think that a major missing component is why does the CAT procedure lead to empirical performance improvements; This has to be substantiated by concrete results, either from a theoretical or an empirical perspective. In either case, the paper can be improved by clearly stating the scope/goal.

- Related work: Please clearly describe the technical innovation of the main theoretical result, and what is new from a technical standpoint.

- Experimental evaluation: This contribution of this part is relatively weak to justify an acceptance in my honest opinion.
    - The datasets + GNN architectures are a bit restrictive, in my opinion. There are many ways to improve the performance of GNNs in practice. I'm not convinced that running the proposed transformation would be the most effective way for practitioners.
    - There are no ablation analysis or robustness checks done here to ensure that results would generalize to different types of empirical applications.

- Page 3: Please include the mathematical definition of a graph being outerplanar.

- Lemma 1: Please include a proof of this result in the appendix. Similarly, the citation to Mitchell (1979) might appear unfamiliar to most TMLR audience (as well as various other citations to the graph-theoretic literature; please check).

- Page 5: Please include some examples for the transformation in Definition 1 to help readers digest this concept.

- Page 6: Likewise, please add more explanations to Definition 2. My concern is that this would be difficult to understand for many of the TMLR audiences.

- Minor: What do you mean by "maximally expressive"? I would like to know what the mathematical definition of this is and I think it would be helpful if this is defined somewhere in the preliminaries.

**Strengths And Weaknesses:**

Strengths:

- The mathematical component of this paper, where the authors introduce a new transformation algorithm for mapping outer planar graphs, appears to be interesting. Plus the empirical evidences are supportive of the utility of this transformation.

- The paper is easy to read for the most part. However, the targeted audience is not clear and the paper would require a major revision at its current form in my opinion.

Weaknesses:

- The empirical evaluations are relatively weak. The baseline models are from five years ago, which are a bit outdated at this point. The coverage of different datasets/applications is not clearly stated, making the implications a bit narrow at this point.

- The connection between the theoretical component (expressivity) and the empirical results is completely missing at this point. This is a major omission in my opinion and needs to be addressed (before the paper can be accepted); I'm thus marking no for claims and evidences partly due to this reason.

---

> ### Author Response · Authors · 2024-09-20
>
> Thank you for your review.
>
> > The empirical evaluations are relatively weak. The baseline models are from five years ago, which are a bit outdated at this point. The coverage of different datasets/applications is not clearly stated, making the implications a bit narrow at this point.
>
> While the baseline models used in the evaluation may be older, it is surprisingly difficult to find datasets in which new architectures beat simple baselines when evaluated in a fair setting. See for example [1] which demonstrated that MPNN baselines when tuned properly beat novel “state-of-the-art” GNN architectures on some datasets. For datasets, we chose molecular datasets that are used as benchmarks in the community (see for example [2,3,4]). As the prevalence of outerplanar graphs in datasets of chemical molecules is the motivation of our work, we focus on such datasets and do not consider other datasets as relevant.
>
>
>
>
> > The connection between the theoretical component (expressivity) and the empirical results is completely missing at this point. This is a major omission in my opinion and needs to be addressed (before the paper can be accepted); I'm thus marking no for claims and evidences partly due to this reason.
>
>
> Please refer to the global reply.
>
>
> > I'm not quite sure what exactly is the problem statement/scope of this paper [..]
>
> Our goal is to design a GNN that performs well on outerplanar graphs from both a theoretical and empirical view. From the point of theory, current GNNs are known to have either limited expressivity or good expressivity but bad runtime complexity. It is thus not known whether there exist GNNs that are both highly expressive and fast on real-world graphs. Our architecture proves that this is possible by achieving the maximum possible expressivity on an important class of graphs in linear time. It is not trivial how such a theoretical result translates to empirical performance. Please also see the global reply.
>
>
> [1] Tönshoff et al., Where did the gap go, TMLR 2024
>
> [2] Cantürk et al., Graph Positional and Structural Encoder, ICML 2024
>
> [3] Bevilacqua et al., Equivariant Subgraph Aggregation Networks, ICLR 22
>
> [4] Bodnar at al., Weisfeiler and Lehman Go Cellular: CW Networks, NeurIPS 2021

---

> > ### Author Response · Authors · 2024-09-20
> >
> > > Related work: Please clearly describe the technical innovation of the main theoretical result, and what is new from a technical standpoint.
> >
> > Previously proposed (deterministic) expressive GNNs are all expensive to run i.e., they do not run in linear time. For example, $k$-GNN [1], $\delta$-$k$-GNN [2], SpeqNets [3],  ESAN [4] and CWN [5] all require at least quadratic runtime. Furthermore, it is often unclear for which type of graphs such GNNs are maximally expressive. In particular, in a recent survey paper leading GNN researchers have identified expressivity results on relevant classes of graphs as an open problem (Challenge II.4 in [6]).  Our theoretical contribution is the proposition of the CAT graph transformation whose properties remedy these problems. First, it runs in linear time and is thus significantly faster than existing approaches (for a runtime comparison please see the response to Reviewer D1E5). Secondly, it is maximally expressive on a highly relevant class of graphs as outerplanar graphs capture most small molecules. Please also see the global reply.
> >
> >
> > > The datasets + GNN architectures are a bit restrictive, in my opinion. There are many ways to improve the performance of GNNs in practice. I'm not convinced that running the proposed transformation would be the most effective way for practitioners.
> >
> > Please see our replies to Weaknesses 1 and 2, as well as the global reply. Note that we do not claim that our proposed method is necessarily the best or the only method that can be applied to outerplanar graphs to make MPNNs maximally expressive *and* yield good empirical performance. We present a method that is sufficient to distinguish all outerplanar graphs via message passing, which was our goal. We are very interested in alternative and/or better ideas and would love to see this problem being explored in other papers.
> >
> > > There are no ablation analysis or robustness checks done here to ensure that results would generalize to different types of empirical applications.
> >
> > In our experiments, we consider classification and regression tasks on molecular graph datasets. As our work is explicitly motivated by the fact that most molecular graphs are outerplanar, we consider generalization to other application domains beyond chemical applications out of scope.
> >
> > > Page 3: Please include the mathematical definition of a graph being outerplanar.
> >
> > The definition we stated “A graph is outerplanar if it can be drawn in the plane without edge crossings and with all nodes belonging to the exterior face” is a standard one. Equivalently, a graph is outerplanar if and only if it does not contain a [subdivision](https://en.wikipedia.org/wiki/Homeomorphism_(graph_theory)) of one of the two graphs $K_4$ or $K_{2,3}$ . We will include the required details in the appendix (such as, full definition of a planar drawing, faces, etc.).
> >
> > > Lemma 1: Please include a proof of this result in the appendix. Similarly, the citation to Mitchell (1979) might appear unfamiliar to most TMLR audience (as well as various other citations to the graph-theoretic literature; please check).
> >
> > Of course we can add the proof from Mitchell [1979] in the appendix. However, it is not common to do so. The paper can be found online and is written well. Thus, we do not see any benefit in summarizing it.
> >
> > > Page 5: Please include some examples for the transformation in Definition 1 to help readers digest this concept.
> >
> > Thank you for this suggestion, we will include additional examples for the CAT* transformation in the appendix.
> >
> >
> > > Page 6: Likewise, please add more explanations to Definition 2. My concern is that this would be difficult to understand for many of the TMLR audiences.
> >
> > The paper already contains an example (Figure 5) and additional examples in the appendix (Figures 8 to 11). We are happy to add further explanations in a revised version.
> >
> >
> > > Minor: What do you mean by "maximally expressive"? I would like to know what the mathematical definition of this is and I think it would be helpful if this is defined somewhere in the preliminaries.
> >
> > Thank you for this suggestion. We will clarify the meaning of a method being maximally expressive in the revised version.
> >
> > A method is maximally expressive for a set of graphs, if it can distinguish all pairs of non-isomorphic graphs of that set (which means it is able to compute different representations for non-isomorphic graphs).
> >
> > [1] Morris et al., Weisfeiler and Leman Go Neural: Higher-order Graph Neural Networks, AAAI 2019
> >
> > [2] Morris et al., Weisfeiler and Leman go sparse: Towards scalable higher-order graph embeddings, NeurIPS 2020
> >
> > [3] Morris et al., SpeqNets: Sparsity-aware Permutation-equivariant Graph Networks, ICML 2022
> >
> > [4] Bevilacqua et al., Equivariant Subgraph Aggregation Networks, ICLR 22
> >
> > [5] Bodnar at al., Weisfeiler and Lehman Go Cellular: CW Networks, NeurIPS 2021
> >
> > [6] Morris et al., Future Directions in the Theory of Graph Machine Learning, ICML 2024

---

### Review · Reviewer_eiRW · 2024-09-10

**Summary Of Contributions:**

This paper proposes a graph transformation called CAT, which enhances the expressiveness of the WL algorithm on outerplanar graphs. This transformation can potentially be utilized in certain types of graph neural networks to improve their performance. The improved performance is verified using standard molecular benchmark datasets.

**Audience:**

Yes

**Claims And Evidence:**

Yes

**Requested Changes:**

See weakness. Additionally, I have several questions that need the authors' clarification.

1. Could the authors verify some fast GNN algorithms, such as FastGCN, which utilize a pruned graph architecture?

2. What is the reasoning behind the lack of significant improvement when applying CAT to GAT?

3. How does the graph structure affect the performance of CAT? Could the authors provide insights into how certain characteristics of the graph-structured data might help determine whether CAT should be implemented?

**Strengths And Weaknesses:**

Strengths:

1. The motivation of the paper is well-written, and the intuition behind the proposed algorithm is well-described.

2. The discussion of the algorithm and corresponding theoretical insights are clear and convincing.

Weakness:

1. The theoretical analysis is incomplete, as it lacks connections to graph neural networks and does not include any optimization or generalization analysis.

2. The experimental setup is unclear. The implementation of CAT in graph neural networks is not clearly described, and the algorithm only improves the performance in part of the graph neural networks.

3. There is no numerical experiments to quantitatively verify the theoretical insights.

---

> ### Author Response · Authors · 2024-09-20
>
> Thank you for your review.
>
> > The theoretical analysis is incomplete, as it lacks connections to graph neural networks and does not include any optimization or generalization analysis.
>
> Previous results on GNNs have shown that suitable architectures allow for parameterizations such that two nodes obtain the same embedding if and only if they have the same Weisfeiler-Leman color [3]. It is commonly assumed that such parameters are also found during the training process if required to solve a certain learning task. Following this assumption, our analysis has direct implications for the expressivity of GNNs. Please refer to the global reply regarding the connection between theoretical and empirical results.
>
> > The experimental setup is unclear. The implementation of CAT in graph neural networks is not clearly described, and the algorithm only improves the performance in part of the graph neural networks.
>
> The experimental setup is described in Section 5 and Appendix D. The implementation of CAT is described in detail in Appendix D. We will clarify both sections in a revised version. Note that CAT itself is a pre-processing of the graphs and can be combined with arbitrary GNNs. Please also refer to the global answer.
>
>
> > There is no numerical experiments to quantitatively verify the theoretical insights.
>
> Please refer to the global answer regarding the connection between the theoretical and empirical results.
>
> > Could the authors verify some fast GNN algorithms, such as FastGCN, which utilize a pruned graph architecture?
>
> While the result of such an experiment would be interesting, we believe it would not fit with the theoretical contribution of our paper. Any architecture that relies on non-determinism, such as the sampling in FastGCN, is not directly captured by the expressivity paradigm. Indeed, it is known that adding random features to graphs makes GNNs maximally expressive [4] (empirically this usually leads to bad predictions). Thus novel expressive GNN architectures usually focus on the deterministic case. Even if FastGCN+CAT achieves strong results it would be unclear if this improvement comes from the expressivity added by CAT or the randomness (and pruning) of FastGCN.
>
>
>
> > What is the reasoning behind the lack of significant improvement when applying CAT to GAT?
> Please refer to the global answer.
>
>
> How does the graph structure affect the performance of CAT? Could the authors provide insights into how certain characteristics of the graph-structured data might help determine whether CAT should be implemented?
> 	 One straight-forward observation from the experimental results is that CAT achieves good results on the ZINC dataset. For this dataset, it is known that being able to count cycles helps with strong predictive performance (see Fig 5 in [1]). As our architecture is able to count cycles this could explain its performance. Besides that, understanding the interplay of expressivity and empirical performance is unfortunately still a major open problem (Challenge III.1 in the positional paper [2]).
>
> [1] Bouritsas et al., Improving Graph Neural Network Expressivity via Subgraph Isomorphism Counting, IEEE Transactions on Pattern Analysis and Machine Intelligence 2022
>
> [2] Morris et al., Future Directions in the Theory of Graph Machine Learning, ICML 2024
>
> [3] Xu, et al. "How powerful are graph neural networks?." ICLR 2019.
>
> [4] Abboud et al., The Surprising Power of Graph Neural Networks with Random Node Initialization, ICLR 2021

---

### Review · Reviewer_D1E5 · 2024-09-11

**Summary Of Contributions:**

The paper proposes a graph transformation algorithm to improve MPNNs maximal expressiveness on outerplanar graphs. The algorithm enables WL to distinguish all non-isomorphic outerplanar graphs, which can be completed in linear time. Experiments and theory show that the algorithm does not essentially harm the graph connectivity. Also, the newly constructed graph has better performances in multiple GNNs on multiple tasks and datasets.

**Audience:**

Yes

**Broader Impact Concerns:**

No broader impact concerns

**Claims And Evidence:**

Yes

**Requested Changes:**

See weakness

**Strengths And Weaknesses:**

Strength:

- The paper is well-motivated to transform the graph structure into a more expressive manner.
- The theoretical analysis proves that graph transformation by CAT improves the expressive of MPNN.
- The experiments are comprehensive on multiple datasets and GNNs

Weakness:

- The proposed method scales up the graph and costs more in computation for downstream GNNs as shown in Table 2. Assuming we have a large enough dataset, the extra computation can be in turn used for more GNN layers or more model parameters which can also lead to better performance. It would be great to compare the computation cost with CAT and under different model sizes.
-  It would be better to introduce the dataset and tasks used in the paper/appendix for background knowledge.
- Need more analysis on the dataset and form of GNN where CAT failed to improve the performance

---

> ### Author Response · Authors · 2024-09-20
>
> Thank you for your review.
>
> > The proposed method scales up the graph and costs more in computation for downstream GNNs as shown in Table 2. Assuming we have a large enough dataset, the extra computation can be in turn used for more GNN layers or more model parameters which can also lead to better performance. It would be great to compare the computation cost with CAT and under different model sizes.
>
> While it is true that for most neural networks increasing the size of the model leads to a better predictive performance, this is not necessarily the case for GNNs. Indeed, adding more layers to a GNN typically leads to worse results. This phenomenon is well studied and is, for example, explained by oversmoothing and oversquashing (see for example [1] or the survey [2]). For the increase in computation cost, Table 2 shows that CAT increases the total runtime by at most 42% over MPNN baselines. We can compare that against other numbers in the literature. Equivariant subgraph aggregation networks [3] are less expressive than our architecture (on outerplanar graphs) and report an over 100% increase in runtime  (Table 11 in [2], note that this only measure training time so unlike our results it does not cover the additional pre-processing). Other highly expressive GNNs have a similar high increase in runtime both (2, 2)-Speq-NETs [3] and CW Networks [4] report an increase of >100% for training time compared to MPNNs.
>
> [1] Di Giovanni et al., On Over-Squashing in Message Passing Neural Networks: The Impact of Width, Depth, and Topology, ICML 2023
>
> [2] Rusche et al., A Survey on Oversmoothing in Graph Neural Networks, arXiv 2023
>
> [3] Bevilacqua et al., Equivariant Subgraph Aggregation Networks, ICLR 22
>
> [3] Morris et al., SpeqNets: Sparsity-aware Permutation-equivariant Graph Networks, ICML 2022
>
> [4] Bodnar at al., Weisfeiler and Lehman Go Cellular: CW Networks, NeurIPS 2021
>
>
>
> > It would be better to introduce the dataset and tasks used in the paper/appendix for background knowledge
>
> We are happy to include a more detailed description of the dataset and tasks used in the appendix.
>
> > Need more analysis on the dataset and form of GNN where CAT failed to improve the performance
>
> Please refer to the global answer.

---

### Author Response · Authors · 2024-09-20
**Global reply**

We thank the reviewers for their valuable feedback and the effort put into the reviews. We will address issues that were mentioned multiple times here, and provide individual point-by-point answers below. We are eager to answer further questions and discuss with the reviewers.

## Contribution and scope of the paper:

Several reviewers raised questions regarding the scope and contribution of our paper. To clarify this, we would like to put our work in the broader context of recent graph learning research. While many papers on graph neural networks mostly focus on empirical improvements, there is a lack of understanding of why and on which graphs certain methods achieve improvements. One step towards answering this question is the analysis of their expressivity, i.e., the ability of methods to distinguish non-isomorphic graphs. Expressivity turned out to be a fruitful concept for the development of theoretically founded novel architectures, although there is a gap between predictive performance and expressivity that is not yet understood [6]. While maximal expressivity for general graphs is unlikely to be achieved in polynomial time (as no polynomial-time graph isomorphism algorithms are known), there are practical, relevant graph classes allowing polynomial-time solutions.

We propose a linear-time graph transformation that allows standard message-passing neural networks to distinguish all pairs of non-isomorphic outerplanar graphs, a class of graphs, which are highly relevant for molecular property prediction. Moreover, our method only linearly increases the graph size and thereby does not change the overall complexity of the learning algorithm. We believe that our contribution is well aligned with current open challenges in graph learning research, as outlined, for example, in this recent position paper [6].

## Datasets/GNN architectures that perform worse with CAT:

Unfortunately, we do not know why CAT does not improve the performance for all datasets and architectures.  As it is currently not completely understood when more expressive GNNs lead to improvements in practice (see above), a conclusive answer cannot be given.

We evaluated our proposed approach on the standard datasets from the ZINC and MOL families. For GAT we can only speculate that the graph attention mechanism does not work well on the graphs transformed by CAT. With GIN and GCN, CAT indeed improved the performance for 7-8 of 10 datasets. It is unclear why CAT did not improve the performance on MOLLIPO and MOLTOX21 but such variations are to be expected in such an empirical evaluation (in particular with the inherent noise in the training of GNNs).


## Connection between theoretical analysis and empirical results:

We acknowledge that the connection between expressivity and empirical predictive performance is not exhaustively explained in our paper. However, analyzing GNNs in terms of their power to distinguish graphs (i.e., their expressivity) is very common, see, e.g., [1-5]. Going beyond that and understanding this relationship is a recent research direction and still an open problem (see [6] Challenge III.1: Understanding the influence of expressiveness and architectural choices on generalization). We consider expressivity analysis as a kind of worst case analysis: Assuming that a model has to distinguish two graphs for a given learning task, maximal expressivity implies that there is no inherent architectural restriction preventing it from doing so. This does, however, not imply that any given trained model will have good performance. Our experiments serve to answer this latter question about empirical performance of our proposed architecture on several given learning tasks.
In a revised version, we will raise this point in a prominent position. A conclusive explanation why more expressive architectures typically lead to improvements in the predictive performance cannot be given within the scope of this article.

[1] Xu, et al. "How powerful are graph neural networks?." ICLR 2019.

[2] Maron, et al. "Provably powerful graph networks." NeurIPS 2019.

[3] Zhang, et al. "Rethinking the expressive power of GNNs ICLR 2023.

[4] Bevilacqua, et al., Equivariant Subgraph Aggregation Networks, ICLR 22

[5] Bodnar, et al., Weisfeiler and Lehman Go Cellular: CW Networks, NeurIPS 2021

[6] Morris, et al. "Future Directions in Foundations of Graph Machine Learning." ICML 2024.

---

### Decision · Action_Editor_e7Qn · 2024-11-13

**Recommendation:** Accept with minor revision

**Comment:**

See the answer to "Claims and Evidence."

**Audience:**

Yes, I believe researchers who study graph learning on molecular graphs might find this paper of interest.

**Claims And Evidence:**

This paper explores the application of message-passing neural networks (MPNNs) to outerplanar graphs, a graph class particularly prevalent in the pharmaceutical field, where molecule structures are often represented as outerplanar graphs. When applying MPNNs to outerplanar graphs, one faces expressivity limitations: MPNNs are generally constrained by the 1-Weisfeiler-Leman (1-WL) algorithm, meaning that any non-isomorphic graphs indistinguishable by 1-WL are mapped to the same embedding in MPNNs (Morris et al., 2019; Xu et al., 2019). To address this, the paper introduces a graph transformation, the Cyclic Adjacency Transform (CAT), which enables the WL algorithm to distinguish all outerplanar graphs. The premise is that, after applying CAT, MPNNs should exhibit enhanced expressive power, allowing them to assign unique embeddings to different outerplanar graphs that would otherwise be indistinguishable under 1-WL.

Reviewers raised concerns about the connection between the 1-WL test and GNNs' expressive capabilities, but I believe the authors sufficiently clarified the connection in their response. They emphasized that while maximal expressivity implies a model architecture is capable of distinguishing graph structures for a given learning task, it does not guarantee optimal performance in practice. Their experiments thus aim to empirically evaluate the proposed architecture's performance across various learning tasks. I recommend the authors revise the paper's introduction to emphasize this connection more prominently.

The experiments demonstrate that applying the Cyclic Adjacency Transform (CAT) as a preprocessing step improves accuracy across several MPNN architectures, including GIN, GCN, and GAT. However, the reviewers expressed concerns that these architectures may be overly simplistic or outdated. While I agree that evaluating state-of-the-art architectures on these datasets would be great, recent papers often use these established models to illustrate the benefits of novel preprocessing techniques (the authors also highlight this point in their response to the reviewers). Overall, I find the empirical claims to be sufficiently well-supported, even when applied to these commonly used baseline architectures.